# Association of Workplace Culture of Health and Employee Emotional Wellbeing

**DOI:** 10.3390/ijerph191912318

**Published:** 2022-09-28

**Authors:** Michele Wolf Marenus, Mary Marzec, Weiyun Chen

**Affiliations:** 1School of Kinesiology, University of Michigan, Ann Arbor, MI 48109, USA; 2Virgin Pulse Institute, Providence, RI 02902, USA

**Keywords:** workplace, culture of health, employee wellbeing, mental health, stress, work engagement, gender, job class

## Abstract

The study aimed to examine associations between workplace culture of health and employee work engagement, stress, and depression. Employees (*n* = 6235) across 16 companies voluntarily completed the Workplace Culture of Health (COH) Scale and provided data including stress, depression, and biometrics through health risk assessments and screening. We used linear regression analysis with COH scores as the independent variable to predict work engagement, stress, and depression. We included age, gender, job class, organization, and biometrics as covariates in the models. The models showed that total COH scores were a significant predictor of employee work engagement (*b* = 0.75, *p* < 0.001), stress (*b* = −0.08, *p* < 0.001), and depression (*b* = 0.08, *p* < 0.001). Job class was also a significant predictor of work engagement (*b* = 2.18, *p* < 0.001), stress (*b* = 0.95, *p* < 0.001), and depression (*b* = 1.03, *p* = 0.02). Gender was a predictor of stress (*b* = −0.32, *p* < 0.001). Overall, findings indicate a strong workplace culture of health is associated with higher work engagement and lower employee stress and depression independent of individual health status. Measuring cultural wellbeing supportiveness can help inform implementation plans for companies to improve the emotional wellbeing of their employees.

## 1. Introduction

Employee emotional health and wellbeing have become a critical concern for organizations today [1]. A 2021 survey found that 76% of U.S. employees reported at least one symptom of a mental health condition, with the most common symptoms being burnout, depressive feelings, and anxious thoughts [2]. Estimates suggest that 28% of US adults report high stress and approximately 8% experience clinical depression [3,4]. It is likely these numbers are under representative due to many not seeking or having access to treatment. Many employers offer workplace health promotion programs to support and improve the health and wellbeing of their employees [5]. Research has shown that establishing a culture of health in the workplace may help facilitate the effectiveness of health promotion initiatives [6,7,8]

Workplace culture of health refers to the influence of the characteristics of the physical and social environment on behaviors and attitudes related to health and wellbeing in the workplace [9,10]. The primary constructs of workplace culture of health include leadership, policies, programs, supervisor support, peer support, and morale. Theories such as the social-ecological model support the influence of culture as a way to target behavior change. The social-ecological model posits that individuals’ behaviors and health outcomes are influenced interactively by individual, interpersonal, community (workplace) and policy level [11]. In the workplace, interpersonal factors such as coworker relationships and supervisor role modeling and community factors such as access to healthy food options or space to be physically active can positively influence employees’ health choices in the workplace [12,13,14]. In the workplace, socioecological approaches move beyond convincing individual employees to make healthier choices. It necessitates the organizations to create an environment that makes choosing healthy behaviors the easy and convenient choice [6]. Many workplace promotion programs address specific individual health-related behaviors such as smoking cessation or nutrition support [5]. These particular programs, however, may only apply to a subset of employees. Influencing culture through interpersonal, community, and policy level factors can reach employees regardless of where they are in their health journey or their participation in workplace health programs [15]. Further, employees’ health status can shift quickly and frequently, and having a culture that supports employees in all stages can also help address these evolving health concerns.

In recent literature, workplace culture of health has been conceptualized and assessed at both the employer and employee level [16,17]. Employer-level measures are completed by a single representative typically involved in the wellness initiative. This singular perspective may not comprehensively assess the organization’s culture. Nevertheless, positive outcomes have been associated with employer-level health culture. Goetzel and colleagues observed that companies with a high internal culture of health had greater appreciation in company stock price when compared to companies with low internal culture of health [18]. A longitudinal study of 21 employers found that improvement in an organization’s culture of health predicted improved employee health risks [19].

Employee-level culture of health measures assesses how the organization supports health and wellbeing from the employee perspective. Previous studies using employee level culture of health measures observed that culture of health is associated with higher levels of job satisfaction and retention [15], work engagement [16], and healthy behaviors such as physical activity and healthy eating [20]. A 2016 on US government employees (*n* = 4703) found that workplace culture of health was negatively correlated with anxiety and depression [21]. Similar results were observed in employees in China, where higher ratings of workplace culture of health were associated with better psychological wellbeing [22]. Domains of workplace culture of health such as leadership and coworkers support have been found to decrease employee health risk [8] and promote positive health behaviors [20]. However, there is only one U.S.-based study that has examined workplace health culture and emotional wellbeing from employees’ perspective [21]. Other studies in this area have measured workplace culture of health from the employer-level, which may not capture a comprehensive perspective [18,19]. In addition, that study did not use a validated research tool to measure workplace culture of health, limiting its generalizability [21]. There is a need to assess workplace health culture using evidence-based measures. It is also unknown the impact of physical health status on ratings of workplace culture of health and its effect on emotional wellbeing.

Further, few studies have examined job class and gender differences in regard to the relationship between workplace culture of health and employee emotional health outcomes. A study with employees (*n* = 880) at a Korean life insurance company found that job classification moderated the relationship between job satisfaction (which was associated with culture of health), and job retention, where supervisors are more likely to stay in their current position if they are satisfied with their jobs than non-supervisors. Another study showed no differences by gender in workplace culture of health scores [23]. However, research has shown that there are differences in stress [24] and depressive symptoms [25] by gender, warranting further exploration in this area. Understanding differences in gender and job class outcomes can help organizations understand the impact of cultural supports within the complex social arena of workplace environments.

While many studies have examined the relationship between workplace culture of health and employee health outcomes, few studies have looked specifically at emotional wellbeing. In addition, existing studies have not taken into account the existing physical health status of employees in that relationship, and examined the way gender and job class may play a role in the effects of workplace culture on wellbeing. Further, there is a need for studies to use a validated metric to assess workplace culture of health from the employee perspective across organizations.

The purpose of this study was twofold: (1) to determine if there are differences by gender and job class in workplace culture of health scores, work engagement, stress, and depression, and (2) to examine associations between workplace culture of health and work engagement, stress, and depression while controlling for age, gender, job class, and biometric data in a large sample of working adults in the U.S. We hypothesized that there would be gender and job class differences by employee health outcome. In addition, we hypothesized that there would be a positive relationship between workplace health culture and employee health outcomes, even while controlling of individual level factors. Results from this study can help inform the design and implementation of emotional health interventions and show evidence of the benefits of taking a culture of health approach to workplace wellbeing programs. Further, this research can help understand how community-level factors may influence individual outcomes to advance our knowledge of the social ecological model in the field of workplace health promotion.

## 2. Methods

### 2.1. Participants and Data Collection

This study is based on a sample from the Virgin Pulse’s database of workplace culture of health assessment. The culture assessment was administered via Virgin Pulse online platform to U.S. based employees of consenting organizations. Eighteen organizations expressed interest in participating and were given summary reports and wellbeing strategy recommendations. The organizations were from a variety of industries, including higher education, health care, real estate, technology, non-profits, financial services, hospitality, manufacturing, utility, and retail. Each organization helped facilitate employees to take the assessments using the Virgin Pulse platform and via email. Assessments were launched between December 2018 and November 2019 and were available to employees for approximately one month during that date range. Employee participation was voluntary. Prior to completing the survey, all participants were informed and consented to allow the use of the anonymized data for research. There was a participation rate of 14% across organizations.

Participants in this study took the Workplace Culture of Health Scale and gave permission to use matched data from the employees’ health risk assessment data. Health risk assessments (HRA) are an instrument used to collect an employee’s health information that can include biometric data and self-report data, typically provided by an independent 3rd party through one’s employer. Of the 18 organizations that participated, 16 organizations also provided HRA data. The final study sample was comprised of 6235 employees across 16 organizations.

### 2.2. Measures

#### 2.2.1. Workplace Culture of Health (COH) Scale

The Workplace Culture of Health (COH) scale is a 42-item questionnaire designed to measure employee assessment of how their workplace supports health [10]. Participants were asked to rate to what extent they agree or disagree with the statements provided. Responses range from 5 (strongly agree) to 1 (strongly disagree). The scale has been shown to be reliable and valid [10]. This study used a shortened version of the COH scale (24 items). Domains measured in the scale include leadership, policies, programs, supervisor support, coworker support, values, morale, and work engagement. Each subscale was comprised of two items, with the exception of values that has one item and morale that has six items. Scores for each domain and for the total scale are calculated by dividing the total points by the total points possible, then multiplying by 100 to get a summative index score. In accordance with the scale developers, the engagement domain is not included in the total score so it can be analyzed as an outcome variable. Previous research using the full Workplace COH Scale has shown high levels of internal consistency, with Cronbach alpha coefficients ranging from 0.91–0.97. This is consistent with the reliability found in this study for the total score, with a Cronbach alpha of 0.92 [10]. The Cronbach alphas for each subscale were as follows: leadership (α = 0.84), policies (α = 0.67), programs (α = 0.58), supervisor support (α = 0.85), coworker support (α = 0.69), morale (α = 0.83), and work engagement (α = 0.81). Job classification was also an item on the workplace COH scale, where participants were asked to select if they were a “supervisor”, meaning they had direct reports, or “individual contributor”, indicating they had no direct reports. Individual contributors are referred to as non-supervisor employees.

#### 2.2.2. Stress

Stress was measured through a single item self-report question, “in the last month, how do you rate your stress?” Responses ranged from 0 (no stress) to 10 (high stress). This measure was included with the Workplace COH Scale.

#### 2.2.3. Depression

The Patient Health Questionnaire Anxiety (PHQ-2) is a 2-item measure designed for a person to self-rate how frequently they experienced depressive symptoms in the past two weeks [26]. This measure was included in the HRA. The participants were asked to rate how often they have been bothered by the statements provided in the questionnaire within the last two weeks. Items on the PHQ-2 include: (1) “little interest or pleasure in doing things” and (2) feeling down, depressed, or hopeless”. Responses to statements range from never experiencing symptoms (0) to experiencing them every day (3). Total scores were calculated by averaging the scores on each item. Individuals who scored 0 to 2 were given a score of 100, from 3 to 4 were given a score of 50, and 5 and 6 were given a score of 0. Scores of 50 or below are indicators of major depressive disorder. Higher scores for depression indicate a lower risk of depression. Previous research has shown this scale has a strong internal reliability with a Cronbach alpha coefficient of 0.83 [26].

#### 2.2.4. Biometric Screening Data

Validated biometric data was collected through the HRA and included: (1) body mass index (BMI), (2) blood pressure, (3) non-HDL cholesterol, and (4) glucose. Pulse pressure was calculated by subtracting diastolic blood pressure from systolic blood pressure. Pulse pressure was used in the regression models to assist with potential multicollinearity issues due to the relationship between systolic and diastolic blood pressure.

### 2.3. Data Analysis

Descriptive statistics including mean and standard deviations were calculated for each variable for the total sample, by job class, and by gender. To explore the relationships between COH scores and employee health outcome variables, bivariate correlations, independent sample *t*-tests, and regression models were performed. Bivariate correlations were conducted for all continuous variables. A linear regression model was conducted to determine the extent to which work engagement, stress, and depression were associated with COH scores while controlling for gender, job class, age, biometric data, and organization. The categories for gender were male and female. The reference group for gender was female in the regression models. The categories for job class were supervisor and non-supervisor. Non-supervisor was the reference group in the regression models. Organization number was included as a categorical variable, and was used to control for any between group differences. Full results of the model are included in Appendix B. The assumption of normality was checked using skewness and kurtosis, which was found to be within normal range for all variables with the exception of depression scores. Multivariate normality was checked using Q-Q plots and the residuals were found to be normally distributed. Multicollinearity was checked for each regression model using the variance inflation factor (VIF). Multicollinearity for each independent variable was tested using VIF and Tolerance (T) scores. The values of VIF for all independent variables ranged from 1.05–1.4 (<5) and the results of T ranged from 0.73 to 0.95 (>0.01), an indication of no multicollinearity. The assumption of homoscedasticity was checked using plotted residuals and was found having non-constant variance for the models, therefore, robust standard errors were used to address the violation [27]. Autocorrelation was checked using ACF plots and the Durbin-Watson test. The Durbin-Watson test statistics ranged from 1.89–1.94, indicating the model did not violate the assumption of autocorrelation [28]. Due to the fact that there were only minor violations of the assumptions of the generalized linear model and that normal distribution of variables is not an assumption of linear regression, we proceeded with analyzing the data [29]. Subsequently, standardized regression coefficients were analyzed to assess the relative importance of each independent variable individually predicting each dependent variable. In addition, t-tests were run to determine differences in COH scores and primary outcome variables by gender and by job class. All analysis was performed using R [30].

## 3. Results

### 3.1. Participant Characteristics

Table 1 and Table 2 show demographic data and descriptive statistics of each study variable by gender, and by job class. Approximately 64% of the sample identified as female, and approximately 19% classified themselves as a supervisor. The average age for the total sample was 45 years old. Regarding employee perception of workplace culture of health, scores above 80 on the COH total score indicate a “high” health culture, scores between 79 to 65 indicate a “medium” health culture, and scores below 65 is considered “low” [31]. The average COH score was 73.86, falling in the “medium” score range for workplace COH. The average stress score was 5.47 out of 10, with over 24% of the sample reporting a score of 8 or higher. Scores from 1 to 3 indicate low stress, from 4 to 7 moderate stress, and scores above 8 indicate extreme stress level [4]. Regarding depression, scores on the PHQ-2 of 50 or below indicate major depression disorder is likely [26]. Our average score of 96.87 indicated there was little to no average risk of depression in our sample, with approximately 5% of the sample was at medium to high risk for depression.

In terms of biometrics, the overall sample had an average BMI score of 29.2, indicating the sample was considered overweight [32]. With regard to glucose level, less than 100 mg/DL is considered not at risk for diabetes [33]. In our sample the mean glucose level of 96.64 mg/DL is within the normal range. Normal blood pressure is typically 120/80 mmHgm [34], indicating our sample average blood pressure level of 120/77 falls within the normal risk, therefore not at risk of hypertension. With regard to non-HDL cholesterol, an indicator of high cholesterol, numbers below 130 are considered optimal. Our sample had a mean level of 130.14, indicating that the sample is just above the normal range [35]. In summary, the mean scores on biometric data indicated that the sample on average was overweight, but were not at high risk for hypertension, diabetes, or high cholesterol. Please see Appendix A for full descriptive statistics for each variable for the full sample, by gender, and by job class.

### 3.2. Differences by Gender and Job Class

Table 1 and Table 2 display group comparisons in outcome variables by gender and job class. There were significant differences between male and female employees in COH score, engagement, and stress. The results of *t*-tests indicated that male employees rated their workplace COH higher, had higher work engagement scores and reported lower stress (*p* < 0.05), compared to the female counterparts. Additionally, there were significant differences between supervisors and non-supervisor employees in COH scores, engagement, stress, and depression. The results of t-tests indicated that supervisors rated the COH higher, had higher work engagement scores, reported higher stress, and lower depression risk than non-supervisor employees (*p* < 0.05). As seen in Table 1 and Table 2, the effect sizes for the comparison by gender in COH scores, engagement, stress, and depression were less than 0.2, indicating that the difference between groups was small, although significant. For job class, there was a small effect size for COH and depression scores. However, there was a small to medium effect size difference in stress (*d* = 0.27) and work engagement (*d* = 0.24) by job class.

### 3.3. Correlations

Figure 1 displays a heat map of the bivariate correlation coefficients between all study variables, including subscales of the COH scale. Darker colors indicate a higher correlation between the variables. All correlations were significant at *p* < 0.001. All subscales of COH were correlated with higher work engagement, lower stress, and lower depression. 

### 3.4. Regression Models Predicting Workplace COH

Table 3 displays the results of the linear regression models, where COH scores is a predictor variable for (A) engagement, (B) stress, and (C) depression while controlling for gender, age, job class, organization, and biometrics. COH score was a significant predictor of engagement (*β* = 0.71, *p* < 0.001), stress (*β* = −0.4, *p* < 0.001), and depression (*β* = 0.08, *p* < 0.001). The results indicated that higher COH scores were associated with increased work engagement, lower stress, and lower depression while controlling for gender, age, job class, organization, and biometric data.

Results of the linear regression also show that job class was a significant predictor of work engagement (*β* = 0.16, *p* < 0.001), stress (*β* = 0.37, *p* < 0.001), and depression (*β* = 0.07, *p* = 0.02), where being a supervisor is associated with increased work engagement, higher stress, and lower depression risk. Age was also a predictor of stress (*β* = −0.04, *p* < 0.001) and depression (*β* = 0.05, *p* < 0.001), where an increase in age was associated with lower stress and depression. Gender was a predictor of stress (β = −0.12, *p* < 0.001), where being female is associated with higher stress. In the depression model, BMI (*β* = −0.04, *p* < 0.001) and glucose (*β* = −0.06, *p* < 0.001) were also predictors, where lower BMI and lower glucose levels were associated with lower risk for depression. Appendix B contains the regression tables with data on all of the covariates included in the model.

## 4. Discussion

This study aimed to explore the employee outcomes associated with a workplace culture of health. We saw that higher culture of health scores were correlated with higher work engagement, lower stress, and lower depression. The models indicated that culture of health was a significant predictor of engagement, stress, and depression while controlling for gender, age, job class, organization, and biometric data.

Our analysis revealed gender differences in employee outcome variables, where male employees rated the culture of health higher, reported high work engagement scores, and had lower stress than females. Research has shown that there are gender differences in stress [24] and depression [25], where females experience greater levels of stress and depression. Previous studies have found that males have scored slightly higher than females on the Utrecht Work Engagement Scale, but the differences lack practical significance [36]. These are consistent with our findings by gender, where we observed males reporting significantly higher work engagement scores and lower stress. Although our results showed a significant difference between employee engagement scores by gender, the effect size of this difference was small in magnitude. In addition, gender did not predict engagement after controlling for covariates in the model. Gender was also a significant predictor of stress even when controlling for other variables in the model. We also observed that males reported higher culture of health scores, inconsistent with previous research by Kwon and Marzec [23] where there were no observed gender differences in regard to workplace culture of health. Prior research has consistently reported differential findings in the experience of males and females in the workplace as it relates to overall culture [37], supporting our findings. In addition, existing gender gaps in pay and family expectations may lead to overall lower perceived workplace support and increased stress [38]. In order to unpack these findings, a qualitative study may be beneficial to explore the gender difference in perceptions of workplace culture support.

Further, we found there were significant differences between supervisors and non-supervisor employees, where supervisors had higher ratings of COH, reported high work engagement scores, higher stress, and lower depression scores, than non-supervisor employees. Our findings suggest that job class may play a role in the perception of workplace culture of health, employee work engagement, stress, and depression. Supervisors reported significantly higher work engagement and stress levels, which may indicate they are at greater risk for job burnout. Still, supervisor employees report higher COH scores. However, Kwon and Marzec [23] found that supervisor experience less stress than non-supervisors, which is inconsistent with our findings. A possible explanation for the differences may be culture as the Kwon and Marzec study involved employees from South Korea while this study surveyed a U.S. based sample. It is possible that stress may be underreported by supervisors to a greater extent. More research on the differences in health and wellbeing outcomes by job class is warranted due to the influence that supervisory employees may have on non-supervisor employees.

Our findings of the relationship between workplace culture of health and work engagement are consistent with results from other studies examining workplace culture of health. A study by Nekula & Koob [16] on 172 employees in Germany found that health culture was a significant predictor of work engagement. Another study of 149 community college employees found that organizational culture of health was a significant predictor of employee engagement [39]. Further, in our study, higher culture of health scores was associated with lower stress and depression scores, consistent with results from previous studies. Kwon and Marzec [23] and Pahn and Yang [40] both observed significant associations between workplace culture of health and occupational stress in their respective studies. With regard to mental health, Jia and colleagues [22] found that workplace culture of health was positively related to mental health and happiness. Laing & Jones [21] also found that workplace culture of health was negatively correlated with depression in a study of 4703 state employees.

There are several strengths of this study. First, this study analyzed one of the most extensive datasets with information on workplace culture of health and employee health outcomes from the employee perspective. Many studies with large sample sizes measure workplace culture of health through a checklist or questions answered by a single company representative [18,19]. Assessment of health culture by a single representative may be biased due to the wide variety of experiences of any given employee at an organization. Second, this study used a valid and reliable measure designed to assess the physical and social environment for supporting health in the workplace from the employee perspective [10]. Other studies that examine workplace health from the employee viewpoint have used assessment tools that have not been validated in the literature [41,42,43]. One of the studies that examined the impact of health culture on mental health and presenteeism only used non-validated two items to measure perceived health culture [21]. Finally, this study used a large sample of employees across organizations and controlled for important covariates such as physical health indicators. Examining employees across multiple companies can help us understand how these relationships may exist regardless of location of employment. Our analysis allowed us to control for the influence of the individual organization, therefore supporting the overarching relationships between perceptions of workplace culture of health and employee health and wellbeing outcomes.

This study makes several unique contributions to the field. This study advances our understanding of the influence of the environmental and community level factors on individual-level outcomes, as posited in the social ecological model. Influencing cultural level factors may see effects beyond what can be expected when targeting individual-level behaviors. In addition, this research advances our knowledge on the importance of culture for all employees, not just physically at-risk individuals. While other studies have observed similar findings in the relationship between mental health and workplace culture of health [21], this study is the first to use a validated tool from the employee viewpoint to understand this relationship. Finally, using gender and job class variables can help researchers and practitioners in the design of workplace health promotion programs. Understanding how culture may have a differential impact on employees by job classification can help inform strategies that address gaps that may exist within the hierarchies of an organization.

Increasingly, organizations recognize the significance of emotional health and seek to provide support for employees. Yet, specifically addressing emotional health with sensitivity to privacy can be challenging in the workplace. This work indicates that establishing a culture that supports health and wellbeing impacts emotional wellbeing positively, independent of individual health status. Committing to a culture of health over simply providing a workplace health promotion program represents a more comprehensive approach on the part of the organization. Treating emotional health can be elusive; therefore, improving culture can be an inclusive way to address this concern.

There are several limitations to note as well. First, this dataset has been stripped of identifiers, making it difficult to add context to the types of employees assessed. Second, workplace culture of health was measured at one-time point, which may not fully capture each employee long-term health status and perception of their workplace’s health culture. Cross-sectional data limits the ability to make directional inferences in the relationships between variables. In addition, most of the data (aside from biometric screenings) are self-reported, which can be subject to bias. Finally, there are limitations with regard to measurement. In particular, it is important to note that the stress measure was only one item, and the depression measure was only two-item. While the PHQ-2 has been validated in the literature [26], a more comprehensive measure may yield more precise information. Further, the researchers only had access to the PHQ-2 total score, so we were unable to calculate internal reliability for the sample. In addition, this study uses a shortened version of the Workplace COH Scale, which has not been validated in previous literature. While the overall scale displayed a high reliability, the policies, programs, and coworker subscale observed lower than ideal internal consistency scores (<0.70), limiting the interpretation and generalizability of the subscale. Further research should focus on collecting longitudinal data to gather a more comprehensive picture of employee perceptions of culture, and individual health and wellbeing status over time. Additionally, most organizations in the study had moderate culture of health, according to our measure. Further work with companies exhibiting strong culture of health and associated outcomes would be valuable. Overall, there are a limited number of studies that have examined workplace culture of health and associated employee outcomes. More research in this area is warranted to help contextualize the results observed in this analysis.

## 5. Conclusions

Overall, the results from this analysis indicated that workplace culture of health scores were significant predictor of engagement, stress, and depression while controlling for gender, age, job class, organization, and biometric data. Our findings of differences by gender and job class can help inform the design and implementation strategies of interventions that target employee wellbeing. While supervisors were more engaged at work than non-supervisors, they were also more susceptible to stress. Non-supervisors reported higher depression scores than supervisors. There was also a gender disparity in health outcomes, with male employees reporting lower stress, and more work engagement. Treating mental health concerns can be elusive and improving an organization’s culture can be a way to target emotional wellbeing without being intrusive. This information provides evidence for designing targeted interventions that address the need of individual groups, rather than a one size fits all approach. Employers who take a culture of health approach by providing supports for environmental and social systems in conjunction with individualized programs may see the greatest impact on the health of their employees.

## Figures and Tables

**Figure 1 ijerph-19-12318-f001:**
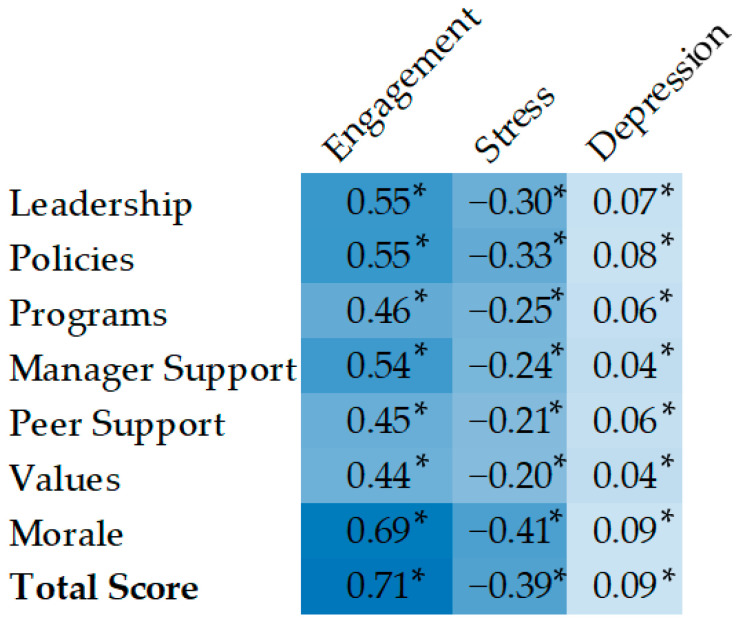
Correlations by Workplace COH subscales and work engagement, stress, and depression. Note: * denotes correlation coefficient was significant at *p* < 0.001.

**Table 1 ijerph-19-12318-t001:** Descriptive statistics and results of *t*-test comparison of employee outcome variable by gender.

	Female		Male					
	(*n* = 4007)		*(n* = 2228)					
	*Mean*	*SD*	*Mean*	*SD*	*t*	*df*	*p*	*d*
COH Score	73.43	13.03	74.64	12.74	−3.58	4691	<0.001	0.09
Engagement	80.91	13.45	81.73	13.96	−2.24	4460	0.025	0.06
Stress	5.61	2.57	5.21	2.63	5.70	4512	<0.001	0.15
Depression	96.64	14.18	97.27	13.12	−1.78	4908	0.07	0.05

Note = Culture of Health (COH).

**Table 2 ijerph-19-12318-t002:** Descriptive statistics and results of *t*-test comparison of employee outcome variable by job class.

	Non-Supervisor		Supervisor					
	(*n* = 5023)		(*n* = 1212)					
	*Mean*	*SD*	*Mean*	*SD*	*t*	*df*	*p*	*d*
COH Score	73.59	13.10	74.99	12.19	−3.58	4691	<0.001	0.09
Engagement	80.56	13.82	83.87	12.49	−2.24	4460	0.025	0.06
Stress	5.33	2.61	6.02	2.46	5.70	4512	<0.001	0.15
Depression	96.62	14.42	97.88	10.92	−1.78	4908	0.07	0.05

Note = Culture of Health (COH).

**Table 3 ijerph-19-12318-t003:** Results of linear regression model with COH score as the main predictor variable.

Variable		*R* ^2^	*F*	*p*	*df*	*β*	*SE*	*t*	*p*
Engagement		0.51	281.1	<0.001	(23, 6211)				
	COH Score					0.71	0.01	77.89	<0.001
	Gender					0.01	0.02	0.37	0.711
	Job Class					0.16	0.02	6.94	<0.001
	Age					−0.01	0.01	−1.02	0.306
	BMI					−0.01	0.01	−1.34	0.181
	Pulse Pressure					0.00	0.01	0.46	0.648
	Glucose					0.00	0.01	−0.44	0.658
	Non HDL					−0.01	0.01	−1.53	0.127
Stress		0.20	65.63	<0.001	(23, 6211)				
	COH Score					−0.40	0.01	−34.43	<0.001
	Gender					−0.12	0.03	−4.72	<0.001
	Job Class					0.37	0.03	12.45	<0.001
	Age					−0.04	0.01	−3.36	<0.001
	BMI					0.00	0.01	−0.30	0.761
	Pulse Pressure					−0.01	0.01	−0.60	0.547
	Glucose					0.00	0.01	−0.40	0.687
	Non HDL					0.00	0.01	0.05	0.957
Depression		0.02	5.25	<0.001	(23, 6211)				
	COH Score					0.08	0.01	5.89	<0.001
	Gender					0.03	0.03	0.95	0.345
	Job Class					0.07	0.03	2.30	0.02
	Age					0.05	0.01	3.49	<0.001
	BMI					−0.04	0.01	−3.03	<0.001
	Pulse Pressure					0.02	0.01	1.15	0.249
	Glucose					−0.06	0.01	−4.48	<0.001
	Non HDL					0.00	0.01	−0.36	0.719

Note. COH = Culture of health, Org = Organization ID number, BMI = Body Mass Index, Non-HDL = Non high-density lipoprotein, *β* = standardized regression coefficient, *b* = unstandardized regression coefficient.

## Data Availability

The data are not publicly available because they contain information that could compromise the privacy of research participants. Please reach out to the corresponding author for more information.

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
