# Peer review of "Association of Workplace Culture of Health and Employee Emotional Wellbeing"

_ijerph, 2022, doi:10.3390/ijerph191912318_

Round 1

Reviewer 1 Report

Thank you for the opportunity to review your manuscript titled Association of Workplace Culture of Health and Employee Emotional Wellbeing. This study explores correlations between workplace culture of health and individual experiences of engagement, stress, and depression. The dataset comprises a large sample of working adults in the US (although this is not specified until the discussion), across multiple companies.

While I appreciate the large sample size and general approach to the study, I do not feel satisfied that the manuscript makes a significant contribution to our knowledge field in its current state. I outline my concerns in further detail below:

1. Not enough explanation of the gap in literature. The authors highlight a few studies that have looked at individual level of WCoH, which appear to report consistent findings. Thus, it’s not clear what the gap in the literature is, nor how this study extends our knowledge. In other words, what does this study mean for our understanding of the culture of workplace health? How might we amend strategies based on this information? The omission of comparisons of WCOH ratings and gender don’t seem to constitute a big enough gap to yield a study of this magnitude.

2. No theoretical model. The authors briefly discuss the socio-ecological model in the introduction and infer it’s use as an underlying framework for the study, but the results nor discussion make no mention the model, nor how theoretical understanding is advanced as a result.

3. The authors should present a much richer discussion of how the study advances the field (i.e., it’s contribution) at the end of the introduction. What are the methodological innovations? Theoretical advancements?

4. Descriptions of the procedure were lacking – the authors should include more detail on how organisations distributed the opportunity to employees (i.e., how was the study promoted), how consent was gained, etc. More demographic information is required as well – from what country, industry were the participants located? What do we know about WCoH in the US compared to other countries?

5. The authors make constant references to “job class” within the manuscript without explaining the term. This should be rectified.

6. On line 28, the authors describe depression and anxiety as symptoms of mental health conditions, when they are in fact the mental health conditions themselves. Please be precise in your terms.

7. The discussion fails to specify how the study adds to the existing literature base – instead, the authors restate how the results align with existing findings and introduce new studies to the mix. I would like to know about the unique contributions that this study makes.

8. Similarly, the authors offer no explanation for why they find gender differences in WCoH ratings, despite this being a key aim for the study. It is not sufficient to say that further study is needed – the authors should examine and propose explanations for why the differences were found in their own data and then identify future steps of enquiry to unpack the findings further.

9. Starting on Line 134 “responses to statements range from never experiencing symptoms (0) to experiencing them every day (3). Individuals who scored 0 to 2 were given a score of 100, from 3 to 4 were given a score of 50, and 5 and 6 were given a score of 0. Scores 3 or above (50 or below) is an indicator of major depressive disorder” – the description is very confusing. Why the conversion to scores out of 100 if the benchmark scores are 0-6? (and participants measure on a scale from 0-3)?

10. On Line 225 there is a strange typo when introducing Table 3.

I hope these comments help improve this piece of work, and I wish the authors good fortune in their future research endeavours.

Reviewer 2 Report

The manuscript entitled "Association of Workplace Culture of Health and Employee Emotional Wellbeing" is related to the important issue in adult life, how workplace culture can predict mental health. Although the article has merit, many problems in the manuscript must be resolved before this work is considered for publication.

1.  The introduction requires improvement. It must clearly state what novelty the manuscript proposes and what the literature gap will fill. It is suggested to present a direct hypothesis based on previous literature. Currently, the authors seem to explore for the first time in the world the associations between workplace culture and mental health, which is surprising considering the rich literature.

2. Measures need improvement. Please add the information, on how many items are included in each subscale of COH. There is a missing reference for the shortened COH, used in this research. The reliability should be reported for each subscale, and the total score of COH for the previous research (with appropriate reference) and the current study sample. Please add the Cronbach's alpha for the PHQ-2 (for the previous and current study).

3. Please add the information on categories of job class and gender, and its coding for the correlation and linear regression, in the "Data analysis" section.  It is currently unclear how to interpret these associations.

4. There are much more assumptions for linear regression than multicollinearity (VIF and tolerance), including the normal distribution of each variable in the model, multivariate normality, homoscedasticity, and autocorrelation. Please examine, report, and comment on these statistics in the "Data analysis" section.

5. What type of 't-test" was performed? There are several possibilities. What statistical software was used in the manuscript? 

6. Results require improvement. Table 1 has no sense in showing the means and standard deviations stratified by gender and job classes, without testing for significant differences. Please, please divide Table 1 into two tables, separately for gender (Table 1) and job classes (Table 2), showing the independent samples Student's t-test statistics, with df, p-value, and Cohen's d, for each variable. Table 3 will be redundant and should be deleted.

7. It makes no sense to show solely unstandardized b statistics when the multiple regression model includes several variables. Table 2 should be rearranged, including standardized beta regression weight and t statistics for each variable in the model. The betas should be also commented on in the text.

8. It is unclear, why the appendix is presented and what hypothesis examines, it since is not commented on in the manuscript at all. Please delete the appendix or comment on its results in detail. What is "Org" in the appendix? 

9. Generally, each table should include a note explaining all abbreviations.

10. Limitations should be extended, discussing the measurement, namely short two-item depression and one-item stress (with unknown reliability).

Reviewer 3 Report

No real suggestions.  Not a surprise to find that the gender variable (female) was significant.   The study would truly be a first of its kind if it could offer explanation regarding the gender variable significance.  That's where some qualitative research would also be useful.  Perhaps another study at another time.

Round 2

Reviewer 1 Report

Thank you for making the changes - the paper is now acceptable to proceed to publication. 

Reviewer 2 Report

The authors significantly improved the manuscript, so the previous comments were well addressed. However, there are some issues that need correction.

1. Still, no PHQ-2 reliability is reported in the current study.

2. Please decide whether you prefer to report significance as a number (exact p-value, e g., 0.029) or asterisks (*, **, ***). Both indicators cannot be included in one table because one of them is redundant. I suggest removing columns with asterisks in Table 1 and Table 2 and all asterisks from Table 3 and Appendix B.

3. The reliability of the three scales of COH (policies, programs, and coworker support) is below standards (< 0.70). Please comment on this weakness of the study in the limitation section.
